# *Trichoderma harzianum* Cellobiohydrolase Thph2 Induces Reactive Oxygen Species-Mediated Resistance Against Southern Corn Leaf Blight in Maize

**DOI:** 10.3390/jof11090629

**Published:** 2025-08-27

**Authors:** Bo Lang, Hongyi Liu, Gaoyue Si, Xifen Zhang, Cheng Zhang, Jing Wang, Jie Chen

**Affiliations:** 1School of Agriculture and Biology, Shanghai Jiao Tong University, Shanghai 200240, China; langbobo2@sjtu.edu.cn (B.L.); liuhongyi@sjtu.edu.cn (H.L.); gaoyuesi@163.com (G.S.); zhangxifen0807@163.com (X.Z.); zhangcheng251@sjtu.edu.cn (C.Z.); 2The State Key Laboratory of Microbial Metabolism, Shanghai Jiao Tong University, Shanghai 200240, China; 3Research Center of Soil Resource Comprehensive Utilization and Ecological Environment in Western Inner Mongolia, Hetao College, Bayannur 015000, China; 4Inner Mongolia Research Institute, Shanghai Jiao Tong University, Linhe District, Bayannur 010052, China

**Keywords:** *Trichoderma harzianum*, cellulase, plant immune response, cellobiohydrolase, reactive oxygen species (ROS)

## Abstract

The pathogenic plant fungus *Bipolaris maydis* is responsible for southern corn leaf blight (SCLB), a widespread agricultural disease that significantly reduces maize yield in various agroecological zones. The present research focuses on characterizing the role of *Trichoderma harzianum* cellobiohydrolase (CBH) Thph2 in induced maize resistance to SCLB by triggering the production of reactive oxygen species (ROS) in leaves. First of all, we demonstrated the potential activities of Thph2 in triggering ROS burst and PDC in a model plant, *Nicotiana benthamiana*. Cell death, ROS burst, and programmed cell death (PCD) were observed in *N. benthamiana* leaves following transient expression of Thph2, indicating its defensive role against *Sclerotinia sclerotiorum* infection. The removal of the signal peptide from Thph2 resulted in the complete loss of the cell death phenotype and the accumulation of reactive oxygen species (ROS), confirming that Thph2 functions as a microbial elicitor that primes host plant immunity through ROS-mediated signaling, thereby inducing systemic resistance (ISR). Furthermore, the Thph2 protein conferred resistance against *B. maydis* in maize, significantly increasing reactive oxygen species (ROS) accumulation (1.5-fold compared to the control) at 48 h post-inoculation (hpi),and leading to the reduction in the lesion area of SCLB by 15.9% at 2 days post-inoculation (dpi). Our results demonstrated that the Thph2 protein markedly enhanced the expression of *lox5*, *aos*, and *hpl* in maize leaves, thereby confirming its function in triggering plant defense mechanisms primarily via the jasmonic acid signaling pathway. This research reveals new molecular mechanisms by which *T. harzianum* enhances plant defense and showcases the biocontrol efficacy of Thph2 against southern corn leaf blight (SCLB).

## 1. Introduction

The plant immune system represents a sophisticated and tightly regulated defense network that provides protection against diverse pathogenic threats, encompassing bacterial, fungal, oomycete, and viral invaders. Contemporary understanding divides the plant immune system into two principal strata: pattern-triggered immunity (PTI) serving as the first line of defense, and effector-triggered immunity (ETI) providing more specific resistance [1,2]. The distinction between PTI and ETI is not as clear-cut as initially believed, as there is significant overlap between the two, even at the receptor level [3,4]. As the primary defense barrier, PTI is triggered when plant pattern-recognition receptors (PRRs) detect pathogen-associated (PAMPs) or microbe-associated molecular patterns (MAMPs) [5]. For these PAMPs/MAMPs [6,7,8,9], upon detection, PTI triggers a cascade of defense mechanisms, which encompass the generation of reactive oxygen species (ROS) [10,11], the deposition of callose, and the activation of genes associated with defense responses.

*Triochoderma* spp. are widely recognized as effective biocontrol agents, with numerous studies demonstrating their capacity to induce plant resistance against diverse pathogens. [7,8,12,13]. The interaction between plant roots and fungi from the genus *Trichoderma* enhances the plant’s immune response, preparing it for potential infections [14,15,16]. In the context of plant immunity, a common trade-off exists between growth and defensive responses. However, certain strains of *Trichoderma* have been shown to enhance plant immunity while simultaneously promoting growth. Initially, plant roots perceive the fungus as a potential threat. During this phase, *Trichoderma* must circumvent the plant’s defense mechanisms to successfully colonize the root epidermis and outer cortex. Concurrently, it is essential [17,18]. A primary mechanism through which *Trichoderma* provides biocontrol against foliar pathogens is by inducing plant resistance via MAMPs or elicitors. These include various hydrolytic enzymes such as chitinase, endogalacturonase, cellulase, xylanase, beta-1,3-glucanase, and proteases, among others [19,20]. For instance, *T. reesei*, which possesses a high concentration of hydrolytic enzymes, can decrease the severity of gray blight by 67.5% to 75.0% compared to pathogen-inoculated controls. Cellulase (EC 3.2.1.4) infiltration from *T. longibrachiatum* into melon (*Cucumis melo* L.) cotyledons triggers crucial defense responses, including hypersensitive reaction-like symptoms. This cascade begins with an oxidative burst occurring three hours after treatment, which then triggers the activation of ethylene and salicylic acid (SA) signaling pathways, leading to the upregulation of peroxidase and chitinase activities [21]. The endogalacturonase (ThPG1) produced by *T. harzianum* serves as an initiator, promoting root colonization by *Trichoderma* and enhancing leaf resistance to *B. cinerea* [5,22]. To date, mechanistic studies of *Trichoderma*-mediated induction of leaf spot resistance in plants remain largely limited to effector characterization and transcriptional profiling of defense-related genes, with limited exploration of downstream signaling pathways or metabolic reprogramming [21,23] Over 20 resistance-inducing effectors have been identified in *Trichoderma* species, including secretory proteins, cell wall-degrading enzymes (e.g., chitinases), phytohormones (e.g., IAA), polysaccharides, and antimicrobial secondary metabolites (e.g., peptaibols) [24].

The cellulase system is characterized by three principal enzymatic activities: endoglucanase (EG), exoglucanase (CBH) [25], and β-glucosidase (BG). CBH includes two isomerases, CBHI and CBHII [26]. CBHI specifically cleaves cellulose chains at their reducing ends, while CBHII targets non-reducing ends, with both enzymes producing cellobiose as the primary product. In filamentous fungi, the secretion level of cellobiohydrolase II (CBHII) is generally lower than that of CBHI, although CBHII exhibits approximately two-fold higher catalytic efficiency toward microcrystalline cellulose degradation compared to CBHI [27]. The cellobiohydrolase (CBH) derived from *Trichoderma* species has been demonstrated to significantly contribute to fungal cell wall decomposition, consequently suppressing pathogenic growth. Nevertheless, the mechanism by which cellobiohydrolase (CBH) elicits plant defense responses—especially via reactive oxygen species (ROS) modulation—remains largely elusive [28].

Southern corn leaf blight (SCLB) is a devastating foliar disease caused by *B. maydis*. This pathogen has a global distribution in maize-producing regions, particularly under warm and humid cultivation conditions. Southern corn leaf blight (SCLB) can lead to significant yield losses in maize [29]. Conventional control measures like chemical fungicides, while effective, incur environmental hazards and are incompatible with sustainable agricultural practices. Consequently, developing sustainable and environmentally benign strategies for southern corn leaf blight (SCLB) management represents a critical agricultural priority. In this study, we aimed to elucidate the role of *T. harzianum* CBHII Thph2 in enhancing maize resistance to SCLB through the induction of ROS-mediated defense responses. Previous studies have revealed that CBHII of *T.harzianum* can systemically induce maize resistance to *Curvularia lunata* leaf spot, but remain unclearly on its role in the induced maize resistance against SCLB.

Therefore, characterizing the role of *T. harzianum* CBHII in conferring SCLB resistance to maize is imperative for sustainable disease management, death (PCD) in plant leaves; and elucidating the biochemical defense mechanisms in maize leaves activated by CBHII (Thph2) in response to SCLB following pre-treatment with CBHII (Thph2).

This study elucidates novel defense mechanisms in maize mediated by *T. harzianum*, with particular emphasis on the Thph2 protein as a potent elicitor of induced systemic resistance (ISR) against SCLB caused by *B. maydis.*

## 2. Materials and Methods

### 2.1. Plant Growth Conditions

The variety of tobacco is *Nicotiana benthamiana*, and the variety of maize is the inbred line B104.

All *N. benthamiana* plants were grown in phytotron chambers maintained at a constant temperature of 22 °C, with a photoperiod of 16 h of light followed by 8 h of darkness.

Maize seeds (B104 inbred line) were sterilized by shaking in 75% ethanol for 15 min, followed by treatment with 2% (*w*/*v*) sodium hypochlorite (NaOCl) (active ingredient) for 7 min. Following sterilization, the seeds were rinsed five times with sterile double-distilled water (ddH_2_O) to ensure complete removal of chemical residues. This sterilization process effectively eliminates surface contaminants from seeds, thereby providing aseptic conditions essential for downstream applications including germination assays and tissue culture experiments. The sterilized seeds were placed on sterilized filter papers (55 × 55 mm) that had been pre-soaked in sterile ddH_2_O. The seeds were germinated for 60 h in an environmentally controlled growth chamber (Ningbo Jiangnan Instrument Factory) (25 ± 1 °C, relative humidity 60 ± 5%) under darkness. Plants were maintained under a 16-h photoperiod (16 h light/8 h dark cycle) at 80% relative humidity. Following germination, seedlings were hydroponically grown in 530 mL tissue culture bottles filled with 180 mL of sterile Hoagland’s nutrient solution, without supplemental aeration. Each bottle contained two seeds, secured with sterile supports to ensure stability.

### 2.2. Biological Materials and Culture Conditions

*T. harzianum* (T30), *S. sclerotiorum*, and *B. maydis* grown on potato dextrose agar (PDA) at a temperature of 28 °C were provided by the Shanghai Jiao Tong University Culture Preservation Center, Shanghai, China. *Agrobacterium tumefaciens* GV3101 and *Pichia pastoris* GS115 were cultured at 28 °C in Luria–Bertani (LB) medium and yeast extract peptone dextrose (YPD) medium, respectively. *Escherichia coli* DH5α was cultivated in LB medium at 37 °C.

### 2.3. RNA-Seq Data Analysis and Transcript Quantification

The signal peptide of Thph2 was predicted using SignalP v5.0. Homologous gene sequences were retrieved from NCBI (https://services.healthtech.dtu.dk/services/SignalP-5.0/), accessed on 1 June 2020. And protein domains were analyzed with InterProScan (https://www.ebi.ac.uk/interpro/search/sequence/), accessed on 1 June 2020.

Transcriptome sequencing was performed by Shanghai Personal Biotechnology Co., Ltd., Shanghai, China.

### 2.4. Validation of Signal Peptide Function

The pSUC2 vector, which contained the sucrose invertase gene (SUC2) but lacked both the ATG initiation codon and native signal peptide, served as the cloning platform for the signal peptide sequence. The resulting construct was then introduced into the YTK12 yeast strain through a transformation process.

Transformants were selected on two distinct media: (i) CMD-W plates (0.67% yeast nitrogen base, 0.1% glucose, 2% sucrose, 2% agar, 0.075% tryptophan-deficient supplement); (ii) YPRAA plates (1% yeast extract, 2% peptone, 2% raffinose, 2 μg/mL antimycin A, 2% agar). Control groups consisted of YTK12 strains transformed with either the empty pSUC2 vector (negative control) or pSUC2-Avr1bSP construct (positive control). The activity of sucrose invertase was measured by the enzymatic transformation of 2,3,5-triphenyl tetrazolium chloride (TTC) into its red formazan derivative, 1,3,5-triphenylformazan (TPF). The Yeast Signal Trap Assay Kit was commercially obtained from Coolaber Science & Technology Co., Ltd., Beijing, China.

### 2.5. Thph2 Function Analysis in N. benthamiana

To evaluate cell death, *A*. *tumefaciens* GV3101 strains containing either PVX: Thph2: HA, PVX: ^Δsp^Thph2: HA (lacking the signal peptide), PVX: eGFP:HA, or PVX: Bax constructs were collected by centrifugation and subsequently washed three times with 10 mM. The bacterial suspensions were subsequently resuspended and normalized to an OD600 of 0.8. The bacterial suspensions were infiltrated into plant leaves using a 1 mL syringe. For subcellular localization studies in *N. benthamiana*, the Thph2: GFP and ^Δsp^Thph2: GFP (signal peptide-deleted) constructs were introduced into *A*. *tumefaciens* strain GV3101. Four-week-old leaves of *N*. *benthamiana* were subjected to vacuum infiltration with a suspension of Agrobacterium tumefaciens (OD600 = 0.8) and were kept at a controlled temperature of 22 °C. At 48 h following infiltration (hpi), epidermal leaf cells were subjected to a 70 mM NaCl treatment for 10–20 min prior to microscopic observation.

### 2.6. Detection of Reactive Oxygen Species (ROS) and Cell Death in Plant Systems

Cell death was assessed by incubating plant tissues in 0.1% (*w*/*v*) Evans blue solution for 16 h, followed by thorough rinsing with distilled water to eliminate any nonspecific staining. Following staining, tissues are rinsed extensively with distilled water to remove unbound dye. Cells with disrupted plasma membranes retain Evans blue, while intact cells remain unstained. The samples are then examined under a bright-field microscope, and images are captured for further analysis. For DAB staining, the leaves were submerged in a freshly prepared DAB solution (1 mg/mL containing 0.05% Tween-20, pH 3.8 adjusted with HCl) and vacuum-infiltrated for 5 min. After 8 h incubation in darkness, samples were heated in 95% ethanol at 65 °C for 15 min to decolorize. Brown polymerization products indicate H_2_O_2_ localization.

To measure ROS burst, leaf discs (1 cm^2^) from *N. benthamiana* were equilibrated in a 96-well plate with 200 μL distilled water for 16 h to minimize wound-induced ROS prior to analysis. For elicitation, leaves were treated with 200 μL of a solution containing: (i) 37.5 μg/mL L0-12, (ii) 25 μg/mL horseradish peroxidase (HRP), and (iii) either flg22 (10 μM), chitin (1 mg/mL), or Thph2 (100 μg/mL) as pathogen-associated molecular patterns (PAMPs). Luminescence signals were quantified at 562 nm wavelength over a 60-min period using a Varioskan LUX multimode microplate reader (SpectraMax i3x, Molecular devices, Shanghai, China). Twelve replicates per condition were measured across three independent trials.

### 2.7. Western Blotting Assay

Protein extraction and quantification:

Total protein was extracted from tobacco leaves using RIPA lysis buffer (Thermo Fisher Scientific, Shanghai, China) supplemented with a protease inhibitor cocktail (Roche, Basel, Switzerland). The lysates were centrifuged at 12,000× *g* for 15 min at 4 °C, and the supernatants were subsequently collected. Protein concentrations were measured using a BCA assay kit (Vazyme Biotech Co., Ltd., Nanjing, China), in accordance with the manufacturer’s guidelines.

SDS-PAGE and immunoblotting:

Equal quantities of protein (20–30 μg per lane) were resolved using 10% or 12% SDS-PAGE and subsequently transferred to PVDF membranes (Millipore, Shanghai, China) via a semi-dry transfer apparatus (Bio-Rad, Hercules, CA, USA). The membranes were incubated with 5% non-fat milk in TBST (Tris-buffered saline with 0.1% Tween-20) for 1 h.

Antibody incubation and detection:

The membranes were incubated overnight at 4 °C with primary antibodies specific to the target protein (anti-HA, anti-GFP, 1:1500 dilution; Abmart, Shanghai, China). Following three washes with TBST, the membranes were incubated for 1 h at room temperature with HRP-conjugated secondary antibodies (1:5000; Santa Cruz Biotechnology, Shanghai, China). Protein bands were detected using enhanced chemiluminescence (ECL) and imaged with a ChemiDoc™ imaging system (Tanon, Shanghai, China).

### 2.8. Expression and Purification of Proteins in Yeast

Heterologous protein expression in *P. pastoris* strain GS115 was induced under optimized culture conditions (30 °C, 250 rpm) following methanol supplementation (0.5% *v/v* daily). Secreted proteins were harvested from clarified supernatant obtained through sequential centrifugation (8000× *g*, 20 min, 4 °C) and 0.22 μm membrane filtration. To analyze intracellular targets, cell pellets were lysed in buffer (50 mM sodium phosphate, 300 mM NaCl, 10 mM imidazole, pH 8.0) and subjected to high-pressure homogenization (EmulsiFlex-C3, Avestin, Ottawa, ON, Canada) with three passes at 15,000 psi.

The eluted protein fractions were concentrated via Amicon^®^ Ultra-15 centrifugal filters (10 kDa MWCO; Millipore) and exchanged into storage buffer (20 mM Tris-HCl, 150 mM NaCl, 10% glycerol, pH 7.5). Protein purity (>95%) was confirmed through SDS-PAGE (12% resolving gel) with Coomassie brilliant blue staining, followed by immunoblotting with anti-His6 monoclonal antibodies (1:1500; Abmart, Shanghai, China). Protein quantification was performed using the Bradford method (Bio-Rad) with BSA standards. After aliquoting, samples were stored at −80 °C for subsequent functional characterization. Three independent biological replicates were processed to ensure reproducibility.

### 2.9. Quantitative Gene Expression Profiling by Reverse Transcription qPCR

Total RNA was extracted from liquid nitrogen-ground tissues (100 mg) using the FastPure Plant RNA Kit (Vazyme, Nanjing, China; RC401). RNA integrity was verified by agarose gel electrophoresis (28S/18S ratio > 1.8) and quantified spectrophotometrically (NanoDrop One, Shanghai, China; A260/A280 = 1.9–2.1).

Genomic DNA contamination was eliminated using a 4× *g* DNA wiper mix (Vazyme, R323). Reverse transcription employed HiScript III Reverse Transcriptase (Vazyme, R323) with oligo (dT)20 primers (1 μg RNA input). qPCR reactions contained ChamQ SYBR Master Mix (Vazyme, Q711) and 0.2 μM gene-specific primers (Appendix A) in 20 μL volumes. Amplification parameters on a LightCycler 96 (Roche): initial denaturation: 95 °C, 30 s, 40 cycles: 95 °C (10 s), 60 °C (30 s), 72 °C (30 s), melt curve: 65–95 °C (0.5 °C increments), the instrument used, Roche LightCycler^®^ 96, is from Roche Switzerland.

Reference genes (18S rRNA for maize; actin for *Trichoderma*) were validated for stable expression (M value < 1.0 via NormFinder, https://seqyuan.shinyapps.io/seqyuan_prosper/, accessed on 1 August 2024). Relative quantification used the 2^−ΔΔCt^ method. Technical triplicates and three biological replicates were analyzed. Statistical significance (*p* < 0.05) was assessed using one-way ANOVA followed by Tukey’s honestly significant difference (HSD) post hoc test in GraphPad Prism version 9.

### 2.10. Statistical Analysis

All data were collected from ≥3 independent biological replicates. Statistically significant differences were determined using one-way analysis of variance (ANOVA) combined with post hoc tests. For comparisons between two sample groups, the Student’s *t*-test was employed, while the Tukey’s honest significant difference (HSD) test was used for other comparisons. A *p*-value of less than 0.05 was considered statistically significant. The biological replicates and significance thresholds for each experiment are indicated in the figure legends. Graphs were generated using Microsoft Excel and GraphPad Prism 9, displaying means ± SEM from replicate experiments.

## 3. Results

### 3.1. Thph2 Can Induce N. benthamiana Cell PCD

Thph2 was identified as GH6 family CBHII (EC 3.2.1.91) based on conserved domain analysis (Figure 1a). The 1,4-beta cellobiohydrolase superfamily is essential for the degradation of plant biomass. To explore the role of secreted proteins more thoroughly, the Thph2 gene was transiently expressed in *N. benthamiana* using *A. tumefaciens* GV3101 to deliver a Potato virus X (PVX) vector. Subsequently, the bacteria were injected into *N. benthamiana* to assess the potential role of Thph2 in either inducing or suppressing plant cell death. Five days post-infiltration, distinctive necrotic lesions developed on *N. benthamiana* leaves, demonstrating Thph2′s ability to trigger localized cell death around the inoculation sites (Figure 1b,c). This cell death response was absent in GFP-expressing control regions. This finding aligns with the plant cell death characteristics noted in *N. benthamiana*. In vivo expression of Thph2 was confirmed by Western blotting with anti-HA antibodies (Figure 1d). Thph2 and Bax infiltration induced 2.4-fold and 2.6-fold increases in electrolyte leakage relative to GFP controls, respectively (Figure 1d). These findings collectively indicate that Thph2 induces cell death in both *N. benthamiana* and maize.

### 3.2. Signal Peptide-Dependent Programmed Cell Death Induction by Thph2

Bioinformatic analysis using Signal P 5.0 predicted a 17-amino acid signal peptide in Thph2 (https://services.healthtech.dtu.dk/services/SignalP-5.0/), accessed on 1 June 2020. The Thph2 signal peptide was evaluated for its secretion function in yeast through the use of an invertase enzyme [30]. The Thph2 signal peptide was cloned into the pSUC2 vector and introduced into YTK12 yeast cells. All transformants carrying either the positive control (pSUC2-Avr1b) or the Thph2 construct (pSUC2-Thph2) demonstrated robust growth on both CMD-W and YPRAA media. pSUC2-Avr1b (the positive control) was able to grow on YPRAA medium, whereas pSUC2 (the negative control) failed to grow on YPRAA plates (Figure 2a). The YTK12 (pSUC2-Avr1b) secretes invertase, which hydrolyzes sucrose into monosaccharides. These monosaccharides react with TTC (2,3,5-triphenyl tetrazolium chloride) to form a red, water-insoluble triphenyltetrazolium chloride. In contrast, pSUC2 does not secrete invertase and therefore does not produce a color reaction with TTC (Figure 2a). The consistency between the results of pSUC2-Thph2 and pSUC2-Avr1b shows that the signal peptide of Thph2 exhibits secretory function.

To validate Thph2 localization and signal peptide function, GFP-tagged full-length Thph2 and its signal peptide-deleted mutant (^ΔSP^Thph2) were expressed in plants under NaCl-induced plasmolysis conditions. Confocal microscopy revealed predominant plasma membrane localization of Thph2: GFP in *N. benthamiana* cells, with limited signal detection in the cell wall and apoplastic space (Figure 2b). To evaluate the cell death-inducing capability of the signal peptide-deleted mutant, ^ΔSP^Thph2 was transiently expressed in *N. benthamiana*. Comparative analysis demonstrated that while full-length Thph2 triggered programmed cell death, the ^ΔSP^Thph2 mutant lacked this activity (Figure 1a). Deletion of the signal peptide completely eliminated Thph2’s capacity to induce programmed cell death, as quantified through electrolyte leakage assays (Figure 1e).

### 3.3. Thph2 Activates Immune Responses in N. Benthamiana

To determine whether Thph2-triggered cell death involves defense responses, reactive oxygen species (ROS) accumulation was measured following transient expression of PVX constructs (GFP control, Thph2, and ^ΔSP^Thph2) in *N. benthamiana*. The ROS burst in leaves expressing the PVX: GFP and PVX: Thph2 was assessed using the elicitors flg22 and chitin. The results indicated that the leaves expressing PVX: Thph2 triggered a ROS burst, which was significantly higher than that observed in the PVX: GFP. At 72 h post-agroinfiltration, DAB staining revealed significant accumulation of reactive oxygen species (ROS) in leaves expressing PVX: Thph2, whereas leaves expressing PVX: GFP or PVX: ^Δsp^Thph2 exhibited minimal ROS accumulation (Figure 3b,c). At 72 h post-protein expression, *S. sclerotiorum* was inoculated to evaluate the induced resistance function. The results showed that the lesion size on leaves expressing PVX: Thph2 was significantly smaller compared to the PVX: GFP group and CK group (Figure 3d,e). Additionally, the half-leaf assay demonstrated that Thph2 could induce systemic resistance in tobacco leaves (Figure 3f,g).

### 3.4. Thph2 Enhances Disease Resistance in N. benthamiana Leaves

We heterologously expressed and purified Thph2 in yeast to enable its application in diverse plants. The purified Thph2 protein was evaluated for its ability to induce reactive oxygen species (ROS) accumulation in *N. benthamiana* leaves, investigating its role in activating plant defense mechanisms. Thph2: His (100 μg/L) induced a ROS burst comparable to the MAMPs flg22 and chitin (Figure 4a). To assess HR-inducing activity and cell necrosis, we infiltrated *N. benthamiana* leaves with different concentrations of Thph2: His, using PBS (negative control) and 00168 (positive control). A pectate lyase, 00168, derived from *Calonectria* spp., is known to strongly induce tobacco cell necrosis. Thph2 at low concentrations did not or only weakly induced necrosis in tobacco, but 100 μg/mL triggered strong necrosis. Thph2 elicited a pronounced necrotic response, as evidenced by distinct necrotic lesions observed through Evans blue staining (Figure 4b). DAB staining revealed that Thph2 at 50 μg/mL effectively triggered ROS accumulation in *N. benthamiana* (Figure 4b). Together, our data establish Thph2 as a fungal proteinaceous elicitor that activates MAMP-triggered immunity, akin to canonical MAMPs like flg22.

### 3.5. Thph2 Can Induce ROS Accumulation in Maize Leaves, Enhance Resistance to SCLB

We evaluated the effect of Thph2 pure protein on maize leaves using two methods: (1) direct injection (PBS control), and (2) surface application with 0.02% diatomaceous earth (PBS and diatomaceous earth controls). The injection experiments demonstrated that Thph2 induces necrosis in maize leaf tissues, concomitant with ROS accumulation (Figure 5a). ROS staining showed significantly increased ROS accumulation in Thph2-treated maize leaves versus controls (Figure 5b).

Maize leaves pre-exposed to Thph2 protein were inoculated with a *B. maydis* suspension (10^5^ spores/mL), then incubated for 24 h at 28 °C in a humidity-controlled chamber (95% RH). Thph2 treatment markedly suppressed lesion expansion, yielding a 15.9% decrease in lesion area relative to controls (*p* < 0.05; Figure 5c). Thph2 significantly increased both SOD and CAT activities, with corresponding upregulation of ZmSOD (2-fold) and ZmCAT (1.5-fold) gene expression relative to controls (*p* < 0.05) (Figure 5d,e). To investigate the JA pathway involvement in Thph2-induced systemic resistance (ISR) in maize [31,32,33], we analyzed the expression of key JA biosynthetic genes (LOX, AOS, and HPL) in Thph2-treated leaves. It was found that the expression levels of these three genes increased by approximately 2.96-fold, 1.71-fold, and 1.79-fold, respectively (Figure 5g). Meanwhile, we analyzed the expression of SA-mediated signaling pathway genes (ZmPR1, ZmPR5, and PAL) in maize [34]. ZmPR1 and PAL expression levels in treated plants were comparable to those in the untreated (control) plants. In contrast, ZmPR5 expression was slightly elevated in treated plants, but the difference was not statistically significant (Figure 5g).

## 4. Discussion

*Trichoderma* spp. are recognized as effective biocontrol agents for various fungal diseases affecting crops, making them an essential element of integrated pest management (IPM) strategies [31,35,36]. For instance, the integrated application of *Trichoderma* and seed treatment with fungicide *T. harzianum* and foliar spray with Tricyclazole 18%, Mancozeb 62% WP and 0.1% at 35 and 50 days after sowing (DAS) resulted in the highest activity of defense-related enzymes, including PO (0.61 U/mg protein), PPO (0.68 U/mg protein), and PAL (177.49 U/mg) protein against SCLB [37]. Nevertheless, identifying multifunctional elicitors derived from *Trichoderma* is essential for advancing next-generation biofungicides. In this study, maize leaves that were pre-treated with Thph2 protein were inoculated with a *B. maydis* suspension at a concentration of 10^5^ spores/mL. The Thph2 treatment significantly inhibited lesion expansion, resulting in a 15.9% reduction in lesion area compared to the control group (*p* < 0.05; Figure 5c). Our earlier work has demonstrated the induced resistance potentials of cellulase from *T. harzianum* for controlling the *Curvularia* leaf spot and *Fusarium* stalk rot [10]. This is consistent with the results we have obtained. Hermosa et al. also demonstrated that *Trichoderma* spp. produce a range of hydrolytic enzymes and secondary metabolites that act as elicitors, triggering plant defense responses such as the production of reactive oxygen species (ROS) and the upregulation of defense-related genes [38]. Thph2 can induce a ROS burst (Figure 4a), which demonstrates this point. Hydrolytic enzymes often exhibit diverse functions, for instance, xylanase (EIX), a well-characterized MAMP produced by *Trichoderma* (a PGPF), triggers immune signaling in host plants. It triggers defensive responses in tobacco (*N. tabacum*) and tomato (*Solanum lycopersicum*), rendering it a valuable resource for research on plant immunity [39].

So far, limited studies have investigated the role of *T. harzianum*-secreted cellulase in induced resistance against southern corn leaf blight (SCLB). We previously found that Thph2 (cellobiohydrolase II) promotes Trichoderma root colonization, systemically activating defense genes (*AOS*, *LOX5*, *HPL*, *OPR1*) and conferring SCLB resistance [19]. This study further demonstrated that thph2 functions as an elicitor capable of directly interacting with maize leaves to induce ROS burst, thereby enhancing host resistance in both *N. benthamiana* against *S. sclerotiorum* and maize (Zea mays) against southern corn leaf blight (SCLB), Collectively, our data suggest that Thph2 functions as a conserved elicitor of PTI, enabling broad-spectrum disease resistance in phylogenetically distant plant species.

The scarcity of cellulose in pathogenic fungal cell walls suggests that CBHII’s primary role lies in triggering plant immune responses rather than exerting direct fungicidal effects. The infiltration of *T. longibrachiatum*-derived cellulase into melon cotyledons activates defense-related pathways, culminating in HR-like cell death phenotypes. An oxidative burst is observed three hours post-treatment, subsequently triggering the activation of ethylene and salicylic acid (SA) signaling pathways, which results in a significant increase in peroxidase and chitinase activities [40]. Cellobiohydrolase (CBH) is a key component of the cellulase enzyme complex, primarily responsible for breaking down plant cellulose into cellobiose. Cellobiose, generated by Thph2-mediated cellulose hydrolysis, is hypothesized to act as a DAMP that triggers pattern-triggered immunity (PTI) in plants [41]; accordingly, it was supposed that CBHII from *T. harzianum* (T30) and cellobiose released from degraded tobacco or maize leaf cellulose might systemically contribute to the induced maize resistance, thus CBHII and cellobiose could be viewed as “double elicitors” which can synergistically induce plant resistance [42].

The role of reactive oxygen species (ROS) in plants is dose-dependent. At moderate levels, ROS act as antimicrobial agents and signaling molecules to activate immune responses. However, excessive ROS accumulation causes oxidative damage to cellular membranes, resulting in electrolyte leakage and cytotoxicity. Our study demonstrated that either transient expression of Thph2 in tobacco leaves or treatment with purified Thph2 protein induced a ROS burst, which played a primary role in activating plant immune responses and subsequently inhibiting *S. sclerotiorum* infection. Similar work has also reported that Endocellulase (EG1) isolated from Rhizoctonia solani has a function to induce the accumulation of reactive oxygen species (ROS) and resistance in maize [43]. Our preliminary findings indicate that the germin-like protein (ZmGLP) in maize roots recognizes Thph2 derived from *T. harzianum* strain T30, subsequently triggering systemic induction of jasmonic acid (JA)-associated gene expression in leaves [10]. However, a critical unanswered question is whether Thph2’s elicitor activity in leaves requires direct interaction with a ZmGLP homolog, analogous to the root recognition system, to activate foliar immunity. Thph2 as a hydrolytic enzyme is able to degrade plant cell wall cellulose, which would improve *Trichoderma* colonization on plant roots and leaves [19]. Kubicek et al. demonstrated that cellulases, including cellobiohydrolases (CBHs), are essential for plant cell wall degradation, enabling fungi to access nutrients and effectively colonize host tissues [44,45,46]. Our results corroborated these findings, showing that the CBHII Thph2 from *T. harzianum* facilitates colonization of both maize roots and leaves. This colonization process may enhance the entry of Thph2 and other fungal elicitors into maize leaves, where they could interact with putative pattern recognition receptors (PRRs) to modulate maize immunity against SCLB and other plant diseases.

Collectively, our results elucidate a previously uncharacterized mechanism by which Thph2 from *T. harzianum* elicits ROS-dependent plant immunity, establishing its potential as an eco-friendly substitute for synthetic fungicides in agricultural practice. Future research should prioritize elucidating the tripartite interaction mechanisms between Thph2 (a *T. harzianum*-derived elicitor), *B.maydis*, and maize leaf tissues. Understanding these molecular dialogues will be critical for developing sustainable ISR (induced systemic resistance) strategies against SCLB.

## Figures and Tables

**Figure 1 jof-11-00629-f001:**
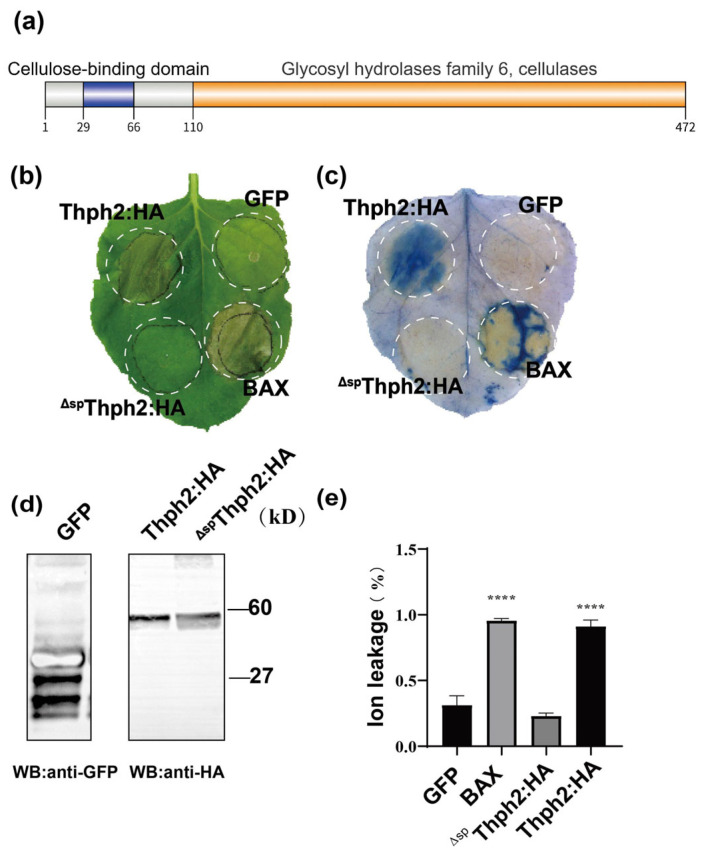
Induction of cell death by Thph2 in *N. benthamiana*. (**a**) The domain diagram of Thph2. (**b**) Induction of cell death in *N. benthamiana* leaves. Thph2 was transiently expressed in the leaves of *N. benthamiana* using the Agrobacterium-mediated system, which included constructs PVX: Bax, PVX: GFP, PVX: Thph2 and PVX: ^ΔSP^Thph2. The same *N. benthamiana* leaf sample shown in (**b**) was stained with Evans blue to visualize cell death morphology. (**c**) Immunoblot analysis using anti-HA antibodies confirmed expression of both GFP:HA and Thph2: HA fusion proteins. (**d**) Electrolyte leakage assays conducted at 7 days post-infiltration (dpi) quantified membrane integrity in *N. benthamiana* leaves expressing GFP (control), Bax (positive control), and Thph2. (**e**) Electrolyte leakage assays at 7 dpi revealed differential membrane permeability in *N. benthamiana* leaves expressing Thph2: HA, ^ΔSP^Thph2: HA (signal peptide-deleted mutant), GFP (negative control), or Bax (positive control) (*n* ≥ 3 biological replicates). Phenotypic documentation was performed at 5 dpi. Asterisks **** denote statistically significant differences (*p* < 0.05, unpaired two-tailed *t*-test).

**Figure 2 jof-11-00629-f002:**
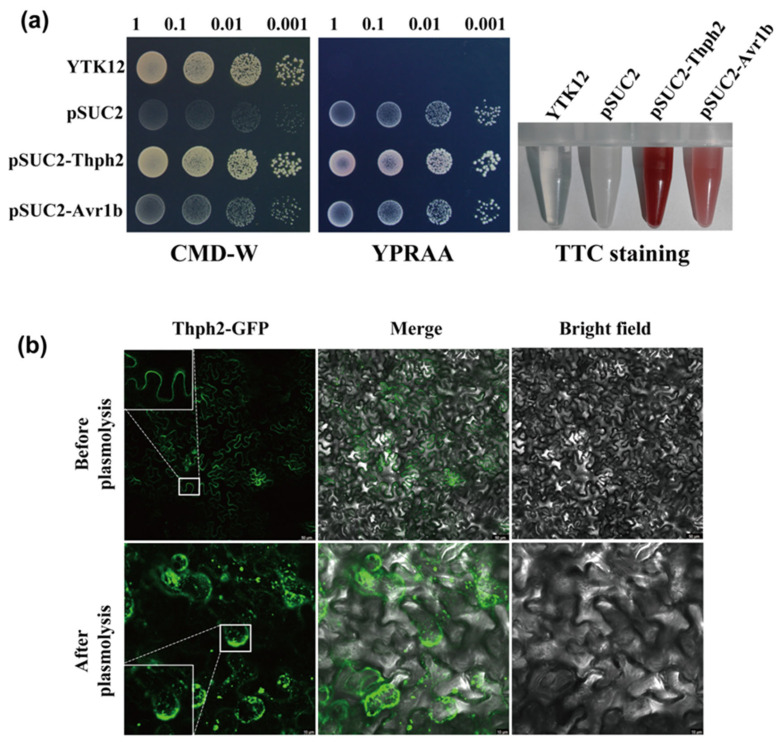
Functional characterization of the Thph2 signal peptide. (**a**) Yeast secretion assay demonstrating signal peptide functionality. YTK12 strains transformed with empty vector (pSUC2), Avr1bSP (positive control), or Thph2SP were plated on CMD-W (selection) and YPRAA (invertase induction) media. Secreted invertase activity was visualized by TTC reduction to red formazan (1,3,5-triphenylformazan). (**b**) Subcellular localization of Thph2 constructs in planta. Full-length Thph2: GFP, signal peptide-deleted ΔspThph2: GFP, and GFP control were co-expressed with plasmolysis marker pBin-Remorin:mCherry in *N. benthamiana* leaves. Epidermal cells were subjected to 70 mM NaCl treatment for plasmolysis induction and imaged by confocal microscopy 48 h post-infiltration (scale bars: 10 μm and 20 μm).

**Figure 3 jof-11-00629-f003:**
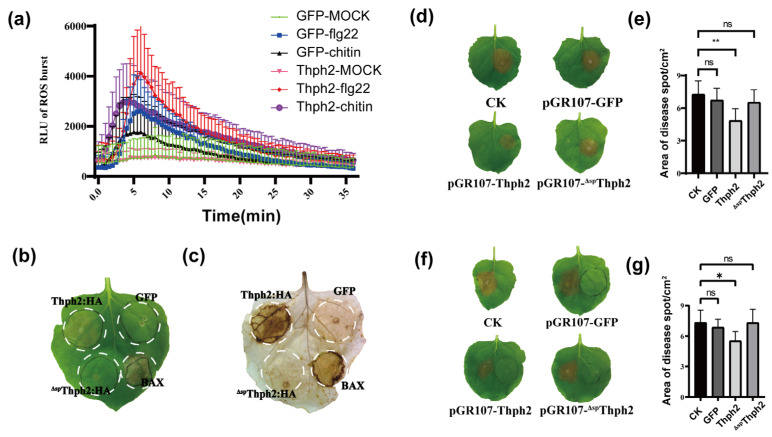
Thph2 elicits robust immune activation in *N. benthamiana*. (**a**) The ROS burst in *N. benthamiana* leaves expressing the PVX: GFP, PVX: Thph2, and PVX: ^Δsp^Thph2 was assessed using 10μM flg22 and 1 mg/mL chitin. Error bars indicate standard deviation (*n* = 12). (**b**,**c**) DAB staining was performed to validate ROS accumulation after expressing the PVX: GFP, PVX: Thph2, and PVX: ^Δsp^Thph2. (**d**,**e**) *N. benthamiana* leaves were treated expressing the PVX: GFP, PVX: Thph2, and PVX: ^Δsp^ Thph2 72 h prior to inoculation with pathogens (*n* = 9). (**f**,**g**) The left side of *N. benthamiana* leaves was treated expressing the PVX: GFP, PVX: Thph2, and PVX: ^Δsp^Thph2 72 h prior to inoculation with pathogen (*n* = 9), and the *S. sclerotiorum* was inoculated on the right side of the leaves. Asterisks denote statistically significant differences (* *p* ≤ 0.05, ** *p* ≤ 0.01; ns: no significance) as determined by ANOVA.

**Figure 4 jof-11-00629-f004:**
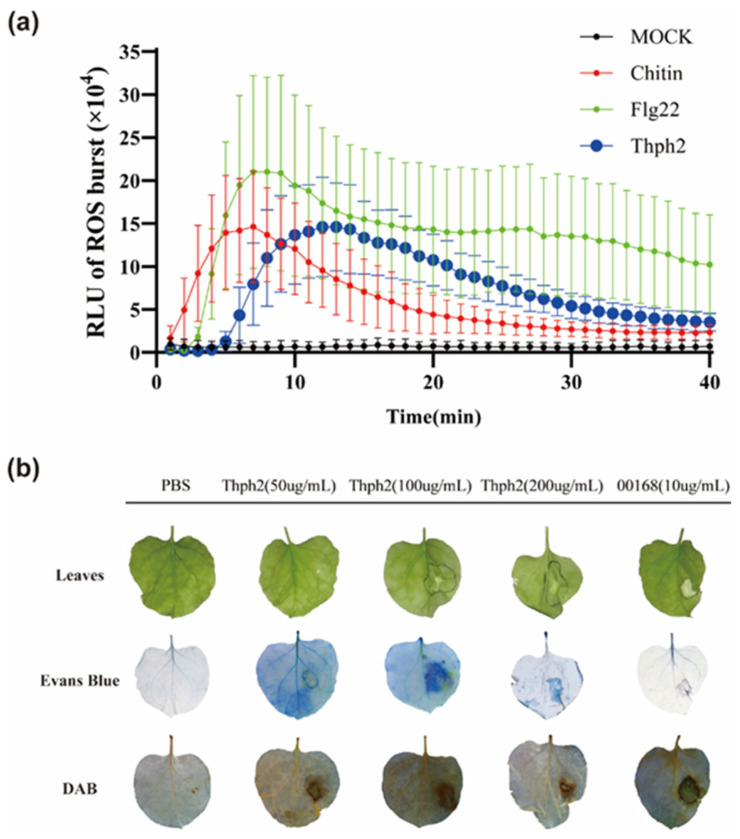
Thph2 induces disease resistance in plants. (**a**) Quantification of total reactive oxygen species (ROS) burst within 45 min post-treatment revealed distinct response patterns in *N. benthamiana* leaves exposed to 100 μg/mL Thph2 protein, flg22 and chitin. Error bars represent the standard deviation (*n* = 12). (**b**) After treating *N. benthamiana* leaves with Thph2 at various concentrations for 72 h, the cell necrosis phenotype was analyzed. Evans blue staining and DAB staining were performed. PBS and 00168 (10 μg/mL) were employed as negative and positive controls, respectively. All experiments were performed with at least three biological replicates.

**Figure 5 jof-11-00629-f005:**
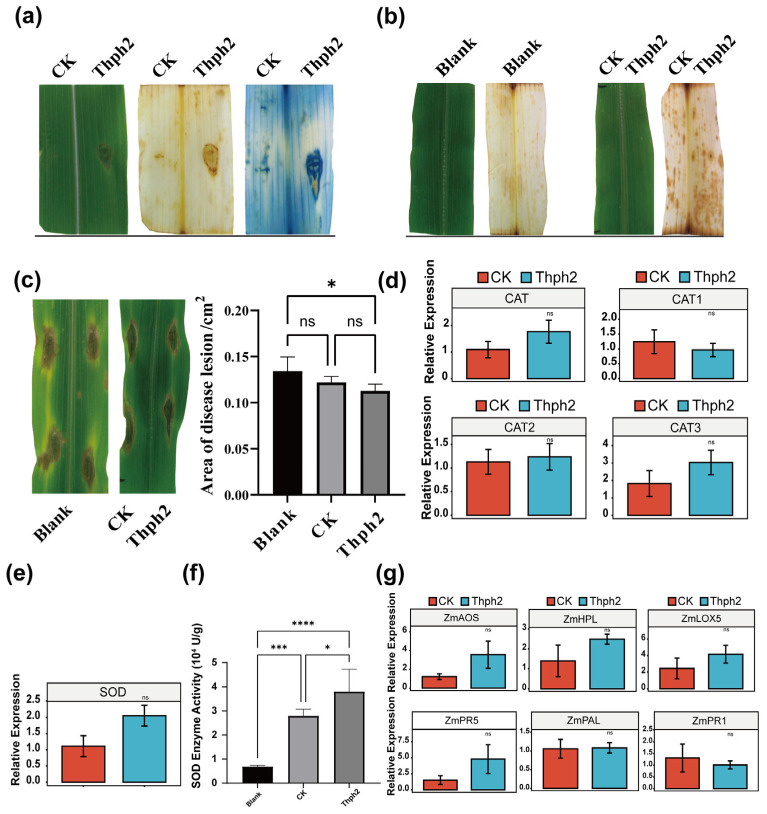
Thph2 induces ROS accumulation and enhances resistance to southern corn leaf blight (SCLB) in maize leaves, (**a**) Necrotic lesions were observed in maize leaves after injection with 200 μg mL/L Thph2 compared to the PBS control (CK), (**b**) ROS accumulation was detected by DAB staining in three groups: untreated (Blank), control (CK: PBS with 0.02% diatomaceous earth), and Thph2-treated (200 μg mL/L with 0.02% diatomaceous earth). Leaves were evenly coated with staining solution using five unidirectional strokes. Quantitative analysis showed significantly higher ROS levels in Thph2-treated leaves (*p* < 0.05). (**c**) After treatments (see Methods), maize leaves were inoculated with *Cochliobolus heterostrophus* * (SCLB pathogen). (**d**) The relative expression levels of different CAT in maize leaves. (**e**) The relative expression levels of SOD in maize leaves. (**f**) Detection of SOD enzyme activity in corn leaves. (**g**) Detection of the expression levels of defense genes in maize leaves. The results represent the means of five replicates for each treatment, with the values indicating the standard error of the mean. Asterisks denote statistically significant differences (* *p* ≤ 0.05, *** *p* ≤ 0.05, **** *p* ≤ 0.01; ns: no significance) as determined by ANOVA.

## Data Availability

The original contributions presented in this study are included in the article and Appendix A. Further inquiries can be directed to the corresponding authors.

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
