# Peer review of "Trichoderma harzianum Cellobiohydrolase Thph2 Induces Reactive Oxygen Species-Mediated Resistance Against Southern Corn Leaf Blight in Maize"

_jof, 2025, doi:10.3390/jof11090629_

Round 1

Reviewer 1 Report

Dear authors!

Thank you for the article provided. It is devoted to the study of the role of cellobiohydrolase Trichoderma harzianum Thph2 in the resistance of maize to pathogenic fungus Bipolaris maydis. The study was conducted using a model system - tobacco plants. I consider the study relevant, since a better understanding of the mechanisms of pathogenesis provides an expanded base for the successful fight against phytopathogens. The article is based on the use of modern molecular genetic and biochemical methods.

There are a number of comments and recommendations, taking into account which will help improve the article, I have listed them in the relevant sections of the review.

After revision, the article can be published in the journal "JoF".

Respectfully Yours, reviewer.

July 24, 2025

1. The abbreviation "PDC" must be deciphered in the Abstract. 2. The keywords must not duplicate the words included in the title of the article. 3. In paragraph "2.1. Plant Growth Conditions" it is necessary to indicate the tobacco cultivation environment. It is necessary to indicate the variety of tobacco and corn. 4. In the Methodological part of plants it is necessary to indicate the strains of microorganisms and where they were obtained. 5. In the Methodological part it is necessary to indicate the manufacturer, headquarters and country in brackets when indicating devices. 6. In paragraph 2.9 it is necessary to provide the sequence of primers. 7. In paragraph 2.10 it is necessary to indicate the criteria by which the sample was checked for normality and the criteria by which the reliability of data differences was checked. 8. The Results chapter should contain only descriptions of your data, and not information from the literature. 9. It is necessary to format the text according to the rules of the journal, make the appropriate alignment (lines 366-386). 10. The graphs in the article are too small. It is impossible to read the labels on the axes and assess the reliability of the differences in data. 11. The Discussion chapter mainly provides literary information, while the authors almost do not describe or explain the results obtained in this study. 12. References must be formatted according to the rules of the journal. It is necessary to increase the citation rate of references over the past 5 years. 13. The membranes presented in the Supplementary file(s) are poorly formatted. As I understand it, the same membrane is presented several times. It is necessary to provide the membrane once and indicate the marker values on it.    

Author Response

Dear Editors,

Thank you for your letter and the reviewer’s comments concerning our manuscript entitled “Trichoderma harzianum Cellobiohydrolase Thph2 Induces Reactive Oxygen Species-Mediated Resistance Against Southern Corn Leaf Blight in Maize (ID:3758312). All the comments are valuable and very helpful for revising and improving our manuscript. We have revised our paper according to the reviewer’s comments. Please find our responses as follows:

Reviewer1:

Question 1: The abbreviation "PDC" must be deciphered in the Abstract.

Response :It has already been revised in the manuscript and marked in red font.

Replace “PCD” with “programmed cell death(PCD)”

Question 2: The keywords must not duplicate the words included in the title of the article.

Response: It has already been revised in the manuscript and marked in red font.

Replace “Trichoderma harzianum; plant defense; cellobiohydrolase; Thph2; reactive oxygen species(ROS); southern corn leaf blight; maize” with “ Trichoderma harzianum; Cellulase; Plant immune response; cellobiohydrolase; reactive oxygen species(ROS)”

Question 3: In paragraph "2.1. Plant Growth Conditions" it is necessary to indicate the tobacco cultivation environment. It is necessary to indicate the variety of tobacco and corn.

Response: The varieties of tobacco and maize have been updated and are marked in red. And tobacco cultivation environment has already been described in the manuscript.

Question 4: In the Methodological part of plants it is necessary to indicate the strains of microorganisms and where they were obtained.

Response :The strains of microorganisms and where they were obtained have made a correction and are marked in red.

Question 5: In the Methodological part it is necessary to indicate the manufacturer, headquarters and country in brackets when indicating devices.

Response: The strains of microorganisms and where they were obtained have made a correction and are marked in red.

Question 6: In paragraph 2.9 it is necessary to provide the sequence of primers.

Response: The sequences of the primers used are presented in the Supplementary Table S1.

Question 7: In paragraph 2.10 it is necessary to indicate the criteria by which the sample was checked for normality and the criteria by which the reliability of data differences was checked

Response : This opinion has been supplemented in the manuscript.

Question 8: The Results chapter should contain only descriptions of your data, and not information from the literature.

Response: “The 1,4-beta cellobiohydrolase superfamily is essential for the degradation of plant biomass. The enzymatic conversion of cellulose to glucose necessitates the coordinated activity of three distinct enzyme classes, the process involves three key enzymes: (i) endo-1,4-β-glucanases (EC 3.2.1.4) cleaving internal β-1,4-glycosidic bonds at random positions; (ii) exo-acting cellobiohydrolases (EC 3.2.1.91) processively liberating cellobiose from chain termini; and (iii) β-glucosidases (EC 3.2.1.21) converting cellobiose into glucose monomers through terminal bond hydrolysis."” has been deleted.

Question 9: It is necessary to format the text according to the rules of the journal, make the appropriate alignment (lines 366-386).

Response: This part has already done it.

Question 10: The graphs in the article are too small. It is impossible to read the labels on the axes and assess the reliability of the differences in data.

Response : The font sizes of axes and assess in Fig3 have been increased.

Question 1·: The Discussion chapter mainly provides literary information, while the authors almost do not describe or explain the results obtained in this study.

Response: It has already been supplemented in the manuscript and marked in red font.

Question 12: References must be formatted according to the rules of the journal. It is necessary to increase the citation rate of references over the past 5 years.

Response: It has already been supplemented in the manuscript and marked in red font.

Question 13:

The membranes presented in the Supplementary file(s) are poorly formatted. As I understand it, the same membrane is presented several times. It is necessary to provide the membrane once and indicate the marker values on it.

Response : It has already been supplemented in the Supplementary file and marked in red font.

We hope these responses and revised manuscript could be acceptable for the journal. Also, we look forward to hearing from you. We would be glad to respond to any further questions and comments that reviewers and editor may have.

Sincerely,

Jie Chen

Professor

Shanghai Jiao Tong University

Reviewer 2 Report

This report was focus on characterizing the role of Trichoderma harzianum cellobiohydrolase (CBH) Thph2 in the induced maize resistance to SCLB through triggering the production of reactive oxygen species (ROS) in leaves. First, the authors has been demonstrated the potential activities of Thph2 in a model plant as Nicotiana benthamiana, but when they try tu due this in maize, the conclutions are not so contundent as in nicotiana model. 

The experiments are well conduced and the techniques used are well chosse, but in the last section they do not observe significant differences between treatment and CK condition, as it was shown in Fig 5, specially about gen expression. Nevestheless, the author conclude based on these results that there are activaction on JA pathway.

I considerer that because there is no significant difference between CK and treatment condition, the authors must not conclude that Thph2 significantly increased both SOD and CAT activities or gen expression.

Author Response

Dear Editors,

Thank you for your letter and the reviewer’s comments concerning our manuscript entitled “Trichoderma harzianum Cellobiohydrolase Thph2 Induces Reactive Oxygen Species-Mediated Resistance Against Southern Corn Leaf Blight in Maize (ID:3758312). All the comments are valuable and very helpful for revising and improving our manuscript. We have revised our paper according to the reviewer’s comments. Please find our responses as follows:

Reviewer2:

Question 1: Fig 3 need improve. Section e and g are very small, so it si to dificult to read it

Response: The new pictures have been replaced.

Question 2: This report was focus on characterizing the role of Trichoderma harzianum cellobiohydrolase (CBH) Thph2 in the induced maize resistance to SCLB through triggering the production of reactive oxygen species (ROS) in leaves. First, the authors has been demonstrated the potential activities of Thph2 in a model plant as Nicotiana benthamiana, but when they try tu due this in maize, the conclutions are not so contundent as in nicotiana model. The experiments are well conduced and the techniques used are well chosse, but in the last section they do not observe significant differences between treatment and CK condition, as it was shown in Fig 5, specially about gen expression.

Response: Since I used the method of applying Thph2 for coating, it may cause certain immune responses and affect the size of the lesions. This experiment has been verified in a previously published article (Trichoderma harzianum Cellulase Gene thph2 Affects Trichoderma Root Colonization and Induces Resistance to Southern Leaf Blight in Maize), except that the vaccination method was different. Previously, different mutant strains of Trichoderma were inoculated at the root level, including wild-type strains, knockout strains and overexpression strains(Fig 1). The same trend was obtained, so we concluded that Thph2 can cause resistance of corn to small spot disease.

Figure 1. Effect of T30:GFP, ∆thph2:GFP, and OEthph2:GFP strains on ISR in maize seedlings to C. heterostrophus and expression of defense-related genes in the leaves of T30:GFP-, ∆thph2:GFP-, and OEthph2:GFP-strain-induced maize plants challenged with C. heterostrophus. (A) Phenotypic map and data analysis of maize leaves inoculated in vitro with different mutant strains. (B) Phenotypic map and data analysis of maize leaves inoculated with different mutant strains by a live inoculation. (C) LOX5, HPL, AOS, and OPR1 gene expression at 36 h after the inoculation of C. heterostrophus. (D) ZmPR1, ZmPR5, and PAL gene expression at 36 h after the inoculation of C. heterostrophus. The results are the means of 5 replicates for each treatment; the value is the standard error of the mean. Different letters above the bars indicate significant differences based on ANOVA (significance: * p ≤ 0.05, **** p ≤ 0.01; ns: no significance).

Question 3: Nevestheless, the author conclude based on these results that there are activaction on JA pathway. I considerer that because there is no significant difference between CK and treatment condition, the authors must not conclude that Thph2 significantly increased both SOD and CAT activities or gen expression.

Response: In previously published  article (Trichoderma harzianum Cellulase Gene thph2 Affects Trichoderma Root Colonization and Induces Resistance to Southern Leaf Blight in Maize), we have demonstrated that Thph2 primarily transmits signals through the JA/ET pathway, leaves(Fig 1) and root(Fig2). Therefore, in this article, although the differences are not very pronounced, it is evident that the expression levels of leaf-related marker genes treated with Thph2 have increased. Thus, we infer that Thph2 mainly transmits signals via the JA/ET pathway.

Additionally, through a review of the literature, we found that an expression level difference of more than 1.5-fold can be considered a significant increase in expression. Therefore, we included the results of SOD and CAT expression levels in our findings.

Figure 2. Thph2 protein induced defense response gene expression in maize roots and leaves. (A) Thph2 protein induced the expression of the defense response genes LOX5, AOS, HPL, and OPR1 in maize roots. (B) Thph2 protein induced the expression of the defense response genes ZmPR1, ZmPR5, and PAL in maize roots. (C) Thph2 protein induced the expression of the defense response genes LOX5, AOS, HPL, and OPR1 in maize leaves. (D) Thph2 protein induced the expression of the defense response genes ZmPR1, ZmPR5, and PAL in maize leaves. The results are the means of 6 replicates for each treatment, and 3 biological replicates were performed.

We hope these responses and revised manuscript could be acceptable for the journal. Also, we look forward to hearing from you. We would be glad to respond to any further questions and comments that reviewers and editor may have.

Sincerely,

Jie Chen

Professor

Shanghai Jiao Tong University

Round 2

Reviewer 2 Report

The authors doing the most of changes that were requested. 

All changes in figure were done.